# Protein Hydrolysates from Fenugreek (*Trigonella foenum graecum*) as Nutraceutical Molecules in Colon Cancer Treatment

**DOI:** 10.3390/nu11040724

**Published:** 2019-03-28

**Authors:** Amine Allaoui, Sonia Gascón, Souhila Benomar, Javier Quero, Jesús Osada, Moncef Nasri, María Jesús Rodríguez-Yoldi, Ahmed Boualga

**Affiliations:** 1Laboratoire de Nutrition Clinique et Métabolique, Faculté des Sciences de la Nature et de la Vie, Université Oran1, 31000 Oran, Algeria; amn.allaoui@gmail.com (A.A.); souhila-13@hotmail.fr (S.B.); 2Departamento de Farmacología y Fisiología, Unidad de Fisiología, Facultad de Veterinaria, Universidad de Zaragoza, CIBERobn (ISCIII), IIS Aragón, IA2, 50013 Zaragoza, Spain; gasconmsonia@hotmail.com (S.G.); javierquero94@gmail.com (J.Q.); 3Departamento de Bioquímica y Biología Molecular y Celular, Facultad de Veterinaria, Universidad de Zaragoza, CIBERobn (ISCIII), IIS Aragón, IA2, 50013 Zaragoza, Spain; josada@unizar.es; 4Laboratoire de Génie Enzymatique et de Microbiologie—Ecole Nationale d’Ingénieurs de Sfax, 3038 Sfax, Tunisia; mon_nasri@yahoo.fr

**Keywords:** fenugreek, protein hydrolysate, antiproliferative, apoptosis, antioxidant, Caco2 cells

## Abstract

The application of plant extracts for therapeutic purposes has been used in traditional medicine since the plants are a source of a great variety of chemical compounds that possess biological activity. Actually, the effect of these extracts on diseases such as cancer is being widely studied. Colorectal adenocarcinoma is one of the main causes of cancer related to death and the second most prevalent carcinoma in Western countries. The aim of this work is to study the possible effect of two fenugreek (*Trigonella foenum graecum*) protein hydrolysates on treatment and progression of colorectal cancer. Fenugreek proteins from seeds were hydrolysed by using two enzymes separately, which are named Purafect and Esperase, and were then tested on differentiated and undifferentiated human colonic adenocarcinoma Caco2/TC7 cells. Both hydrolysates did not affect the growth of differentiated cells, while they caused a decrease in undifferentiated cell proliferation by early apoptosis and cell cycle arrest in phase G1. This was triggered by a mitochondrial membrane permeabilization, cytochrome C release to cytoplasm, and caspase-3 activation. In addition, the hydrolysates of fenugreek proteins displayed antioxidant activity since they reduce the intracellular levels of ROS. These findings suggest that fenugreek protein hydrolysates could be used as nutraceutical molecules in colorectal cancer treatment.

## 1. Introduction

Fenugreek is one of the oldest plants used in traditional medicine. It has been used for a long time due to its beneficial properties in the treatment of wounds, abscesses, arthritis, bronchitis, and digestive disorders [1]. The seeds are the most important and useful part of the fenugreek plant [1]. Many of the functional and medicinal properties of fenugreek are attributed to its chemical composition (20–25% protein, 45–50% dietary fiber, 20–25% mucilaginous soluble fiber, 6–8% fatty acids and essential oils, and 2–5% steroidal saponins) [2].

Belguith-Hadriche, et al. [3] and Subhashini, et al. [4] demonstrated that seed fenugreek extracts are effective against free radical mediated diseases. In addition, Madhava Naidu et al. [5] observed that fenugreek husk, which is more rich in fiber, exhibits an important antioxidant property. However, the proteins of fenugreek seeds, unlike other plants, have been barely investigated.

Legume proteins have become a topic of many studies on health being and certain disease treatments. They are associated with a reduction in the incidence of various cancers, cholesterol, type-2 diabetes, and heart disease [6]. Furthermore, protein hydrolysate has the additional advantage of having improved functional properties as compared to the original protein isolates from which they are prepared. This is due to the release of certain bioactive peptides, which are encoded in the native protein molecule. More recently, potential health-promoting properties of peptides in these hydrolysates have been discovered [7].

The antiproliferative property is among the numerous biological activities attributed to hydrolysates. Effectively, several peptides with anticancer activity have been found in food protein hydrolysates as well as colon antitumor activity of egg yolk proteins or the cytotoxic activity on human colon carcinomas and mouse lymphoma cell lines of hydrophobic peptides extracted from soy [8]. The same findings have been reported in many other studies on *Vicia faba* protein hydrolysate [9], common beans peptides [10], and rice brain peptides [11].

Even if the mechanisms underpinning the antiproliferative effect of the protein hydrolysates is not well established, some hypotheses are proposed. For example, Ortiz-Martinez et al. [12] suggest that the antiproliferative effect on HepG2 cells of peptide fractions isolated from maize albumin hydrolysate is based on the induction of apoptosis due to the decrease of antiapoptotic factors expression. However, Xue et al. [13] reported that a chickpea-derived peptide inhibits the proliferation of breast cancer cells by increasing the p53 expression. Yet, Gao et al. [14] found that peptides derived from soy Vglycin activate the expression levels of pro-apoptotic proteins and caspase-3.

Since colorectal cancer is one of the most commonly diagnosed cancers, and it is strongly influenced by diet [8], the aim of this work has been to study the functional properties of the hydrolysed proteins of fenugreek seeds in relation to the treatment of colon cancer. For this, we have measured the possible antiproliferative and antioxidant effect of these hydrolysates on Caco-2 cells, and determined its mechanism of action.

## 2. Materials and Methods

### 2.1. Fenugreek Protein Hydrolysates Preparation

Fenugreek was purchased from a local spices market in the city of Tiaret (Algeria). Seeds were cleaned, grounded to a fine powder, and defatted in Soxhlet, (Labotech LT-6, Rosdorf, Germany), using *n*-hexane for 12 cycles and their proteins were extracted at an isoelectric point (pH 4.5) according to Boye et al. [15], as detailed previously [16]. The protein isolate was freeze dried and then hydrolyzed.

### 2.2. Preparation of FP Hydrolysates (FPHs)

Two hydrolysates were prepared from fenugreek proteins using Esperase^®^ 0.8L (Sigma Chemical, St. Louis, MO, USA) (pH 9; 50 °C) or Purafect^®^ 2000E (Genencor International, Palo Alto, CA, USA) (pH 10; 50 °C). FP were dissolved in distilled water at a proportion of 5% (*w*/*v*). Mixture pH and temperature were adjusted to optimum enzyme activity prior its incorporation. The enzymes were added to the solution at an enzyme/substrate ratio (E/S) of 5. Once the enzyme added, the mixture pH was maintained constant by a continuous addition of 2N NaOH solution. The degree of hydrolysis (DH) of FP was monitored by using a pH-stat method [17].
$$\mathrm{DH}\;\left(\%\right)=\frac{h}{h_{\mathrm{tot}}}\times 100=\frac{B\;\times N_{B}}{\mathrm{MP}}\times \frac{1}{\alpha }\times \frac{1}{h_{\mathrm{tot}}}\times 100$$
where B is the amount (mL) of NaOH consumed to keep the pH constant during the reaction, N_B_ is the normality of NaOH, MP is the mass of protein (g), and α is the average degree of dissociation of the α-NH_2_ groups released during hydrolysis. h_tot_ is the total number of peptide bonds, which was assumed to be 7.6 meq/g.

Hydrolysis was performed for 5 hours. Afterward, the reaction was stopped by heating the solution at 90 °C for 10 min. Then, the digest was cooled at room temperature and centrifuged at 5000× *g* for 15 min. The obtained hydrolysates: Esperase-fenugreek proteins hydrolysate (EFPH) and Purafect-fenugreek proteins hydrolysate (PFPH), were collected, freeze dried, and then stored at 4 °C.

### 2.3. Hydrolysates Proximate Composition

The protein, moisture, lipids, and ash contents in the freeze-dried fenugreek proteins and proteins hydrolysates were determined by using the AOAC methods [18]. A factor of 6.25 was used to convert the nitrogen value to protein. Minerals were analyzed by using Inductively Coupled Plasma-Optical Emission Spectroscopy (ICP-OES, Perkin Elmer 4300DV, Shelton, CT, USA), after dissolving samples in nitric acid (70%).

### 2.4. Amino Acid Analysis

A total of 50 μL of the sample (1 mg proteins/mL) were first hydrolyzed in a vacuum-sealed glass tube for 24 h at 110 °C in the presence of 6 N HCl and 1% phenol. For tryptophane analysis, samples were hydrolyzed in 4N NaOH, as described by Yust et al. [19]. At the end of hydrolysis, the samples pH was adjusted to 7 and filtered through a 0.45 μm cellulose acetate membrane filter.

The samples were then analyzed by the reversed phase HPLC (Agilent 1100 HPLC system, Wilmington, DE, USA) after automatic precolumn derivatization with a combination of OPA-3MPA (o-phtaldialdehyde-3-mercaptopropionic acid) for primary amino acids and FMOC (9-fluorenylmethylchloroformate) for secondary amino acids, following the manufacturer instructions. The separation was done on a reversed-phase Zorbax Eclipse-AAA column (4.6 × 150 mm, 3.5 μm). The quantification was determined by using norleucine as internal standard. The amino acid composition was expressed as the percent of residues.

### 2.5. Cell Culture

The biological activity of fenugreek protein hydrolysates was evaluated on the human colonic adenocarcinoma Caco2 cell line TC7 clone, provided by Dr. Edith Brot-Laroche (Université Pierre et Marie Curie-Paris 6, UMR S 872, Les Cordeliers, France). Caco2/TC7 cells were maintained in a humidified atmosphere of 5% CO_2_ at 37 °C. Cells (passages 38–41) were grown in Dulbecco’s Modified Eagles Medium (DMEM) (Gibco Invitrogen, Paisley, UK) supplemented with 20% fetal bovine serum, 1% non-essential amino acids, and 1% amphotericin (250 U/mL). The cells were passaged enzymatically with 0.25% trypsin-1 mM EDTA and sub-cultured on 25 cm^2^ plastic flasks at a density of 5 × 10^5^ cells per flask. Culture medium was replaced every three days. Cell confluence (80%) was confirmed by the microscopic observance. Experiments were performed in differentiated cells and in cancerous or undifferentiated cells (24 h post-seeding to prevent cell differentiation).

### 2.6. Cell Treatment and Antiproliferative Property Analysis

EFPH and PFPH were diluted in DMEM to the final concentration of 1 mg/mL. For an antiproliferative experiment, 4 × 10^3^ cells were dispensed into each well of a 96-well plate. The culture medium was then replaced after 24 h with fresh medium (without fetal bovine serum, FBS) containing fenugreek protein hydrolysates, with an exposure time of 24, 48, or 72 h. Untreated cells were taken as a control. The anti-proliferative effect was measured with the sulforhodamine B assay, as described by Sánchez-de-Diego et al. [20]. Cells were fixed with 10% trichloroacetic acid (1 h, 4 °C), washed with distilled water, and stained with 4 g/L of sulforhodamine B (20 min, at room temperature). The plates were then washed with 1% acetic acid (*v*/*v*) to remove the unbound dye. Protein-bound dye was extracted with 10 mM Tris base (pH 10.5). Untreated cells were taken as a control (C).

The same experiment was done with the differentiated cells. Lastly, the results were obtained by measuring absorbance (A) with a scanning multi-well spectrophotometer (SPECTROstar Nano Microplate Reader—BMG LABTECH, Ortenberg, Germany) at a wavelength of 562 nm. The anti-proliferative effect was expressed as a percentage of living cells compared to the control, and calculated as follows:$$\mathrm{Viability}\;(\%)=\frac{{\;A}_{\mathrm{sample}}}{A_{\mathrm{control}}}\times 100$$

### 2.7. Apoptosis Measurement

Undifferentiated Caco2/TC7 cells were exposed for 24 h to 1 mg/mL of EFPH or PFPH, then collected and stained with AnnexinV-FTIC in combination with propidium iodide (PI), as described by Sánchez-de-Diego et al. [20]. Untreated cells were used as a negative control. After incubation, cells were transferred to flow-cytometry tubes and washed twice with temperate phosphate-buffered saline and re-suspended in 100 μL Annexin V binding buffer (10 mM Hepes/NaOH, pH 7.4, 140 mM NaCl, 2.5 mM CaCl_2_). Afterward, 5 μL of the Annexin V-FITC and 5 μL of PI were added to each 100 μL of cell suspension. After incubation for 15 min at room temperature in the dark, 400 μL of Annexin binding buffer were added and analyzed by flow cytometry within one hour. The signal intensity was measured using a BD FACSAria (BD Biosciences, Piscataway, NJ, USA) and analyzed using BD FACSDiva (BD Biosciences, San Jose, CA, USA).

### 2.8. Propidium Iodide Staining of DNA Content and Cell Cycle Analysis

The fenugreek protein hydrolysates treated Caco2/TC7 cells were fixed in 70% ice-cold ethanol and stored at 4 °C for 24 h. After centrifugation (2500 rpm, 5 min), cells were rehydrated in PBS and stained with propidium iodide (PI) solution (50 μg/mL) containing RNase A (100 μg/mL). PI stained cells were analysed for DNA content in a BD FACSArray (BD Biosciences, Piscataway, NJ, USA). The red fluorescence emitted by PI was collected by a 620-nm longer pass filter, as a measure of the amount of DNA-bound PI and displayed on a linear scale. Cell cycle distribution was determined on a linear scale. The results were treated with ModFit LT 3.0 (Verity Software House, Topsham, ME, USA) [20].

### 2.9. Mitochondrial Membrane Potential Assay by Flow Cytometry

Caco2/TC7 cells were plated in 25 cm^2^ flask at a density of 3 × 10^5^ cells per flask and incubated for 24 h under standard cell culture conditions. Afterward, cells were treated with 1 mg/mL of fenugreek hydrolysates and incubated for 24 h. The control cells were incubated with a new medium without treatment and without FBS. Then, cells were washed twice with temperate PBS and re-suspended in temperate PBS at a concentration of 1 × 10^6^ cells/mL. Later, 5 μL of 10 μM cationic dye 1,1′,3,3,3′-hexamethylindodicarbo-cyanine iodide (DiIC1) were added to each sample and the cells were incubated 15 min at 37 °C, 5% CO_2_. After the incubation period, 400 μL of PBS were added to each tube and fluorescence was analyzed by flow cytometry using a BD FACSArray equipped with an argon ion laser. Excitation and emission settings were 633 and 658 nm, respectively.

### 2.10. Determination of Caspase 3 and Cytochrome C

Caco2/TC7 cells were plated in a 25 cm^2^ flask at a density of 3 × 10^5^ cells per flask and incubated for 24 h under standard cell culture conditions. Then, 1 mg/mL fenugreek hydrolysate solution was added to the flask and incubated for 24 h.

The caspase-3 analysis were studied as previously described by Sánchez-de-Diego et al. [20]. The cells were fixed in 0.01% formaldehyde for 15 min and centrifuged for 5 min at 300× *g*. Then, the pellet was suspended in 100 μL digitonin lysis buffer (50 mg/mL digitonin, 100 mM KCl, in 1× PBS) and incubated for 15 min in the dark at room temperature (RT). After incubation, cells were washed with 2 mL of PBS containing 0.1% digitonin and centrifuged at 300× *g* for 5 min. The supernatant was discarded and the pellet was re-suspended in 200 μL of PBS containing 0.1% digitonin. In addition, 2 μL of diluted caspase-3 antibody (Novus Biologicals, Abingdon, UK) were added to each sample and the resultant mix was incubated for 1 hour. After incubation, cells were centrifuged at 500× *g*, for 5 min at room temperature, and washed twice with PBS. Lastly, the cells were re-suspended in 400 μL of PBS. Fluorescence was analyzed by flow cytometry (Ex: 494 nm, Em: 520 nm) using a BD FACSArray.

Cells with liberated cytochrome C were analyzed according to Christensen et al. [21] with slight modifications [20]. Cells were initially resuspended thoroughly in 100 µL digitonin permeabilization buffer (50 µg/mL digitonin; 100 mM KCl; in 1× PBS) followed by incubation for 5 min at room temperature. This was followed by fixing the cells with 100 µL of 4% paraformaldehyde (PFA) in PBS for 30 min. Centrifugation (500× *g*, 5 min) was carried out to remove PFA and cells were washed once with 100 µL 1× PBS. Cells were then incubated with 100 µL blocking buffer (3% bovine serum albumin, 0.05% saponin, in 1× PBS) for 15 min at room temperature. Afterward, 2 µL of diluted cytochrome C antibody 7H8-2C12 (Novus Biologicals, Abingdon, UK) was incubated with cells for 1 h. Cells were washed twice with 1× PBS, then re-suspended in 400 µL of blocking buffer, and samples were analyzed by flow cytometry (Ex: 488 nm, Em: 575 nm) in BD FACSArray.

### 2.11. Intracellular Levels of Reactive Oxygen Species (ROS)

The cells were seeded in 96-wells plate at a density of 4 × 10^3^ cells/well. The intracellular level of ROS was assessed using the dichlorofluorescein assay [22]. Caco2/TC7 cells were cultured for 24 h before oxidative stress induction, and then incubated with 100 μL of serum-free culture media with 1 mg/mL of EFPH or PFPH for 24 h. After that, the medium was removed, cells were washed twice with phosphate buffered saline, and incubated for 1 h with 100 μL of 20 μM 2′,7′-dichlorofluorescein diacetate (DCFH-DA) in PBS at 37 °C. After this period, cells were washed and re-suspended in PBS supplemented with 20 mM or 500 µM H_2_O_2_. The formation of the fluorescence oxidized derivative of DCF was monitored at an emission wavelength of 535 nm and an excitation wavelength of 485 nm in a multiplate reader. A measure at time “zero” was performed, cells were then incubated at 37 °C in the multiplate reader, and generation of fluorescence was measured after 20 min. ROS levels were expressed as a percentage of fluorescence (*f*) compared to the control, and reported using the following formula.
$$\mathrm{ROS}\;\mathrm{levels}\;(\%)=\frac{f_{\mathrm{sample}}}{f_{\mathrm{Control}}}\times 100\times \frac{100}{\mathrm{Viability}}$$

### 2.12. Thioredoxin Reductase 1 (TrxR1) Activity Assay

Undifferentiated cells were seeded in a 96-well plate with different protein hydrolysates for 24 h. The cells were then lysed (5 M NaCl, 1 M Tris-HCl pH 8.0, 0.5 M EDTA pH 8.0, SDS 10%, miliQ water) and incubated in a shaking motion for 20 min. After the incubation time, 25 μL of the reaction mixture (500 μL PBS pH 7.0, 80 μL, 100 mM EDTA pH 7.5, 20 μL 0.05% BSA, 100 μL 20 mM NADPH, 300 μL H_2_O) were added to each well. Lastly, the reaction was started by adding 25 μL of 20 mM DTNB in pure ethanol. The absorbance increase was followed at 405 nm every minute for 6 min. Wells with TrxR1 inhibitor (auranofin) were measured in the same conditions to subtract the unspecific activity [20]. Cell protein contents were calculated by the Bradford method [23]. The result is expressed as a percentage of TxrR1 activity of treated cells compared to the TxrR1 activity of C cells.

### 2.13. Statistical Analysis

Data are presented as mean ±SD. Data were subjected to one-way ANOVA and the LSD-Fisher post hoc test. Differences were considered significant at *p* ≤ 0.05.

## 3. Results

### 3.1. Kinetic and Degree of Hydrolysis

The hydrolysis curve of fenugreek proteins, illustrated in Figure 1, showed a first fast reaction kinetics characterized by an initial rapid phase (during the first 60 min for Esperase and the first 15 min for Purafect). At the end of the hydrolysis reaction, the DHs of the protein isolate were 9% with Purafect and 19% with Esperase.

### 3.2. Chemical and Amino Acids Composition of FP and FPHs

Since the properties of protein hydrolysates depend strongly on their composition, the physicochemical composition of fenugreek protein hydrolysates was first analyzed. The proximate composition of EFPH and PFPH and their amino acid composition are shown in Table 1. Protein and lipids contents in EFPH were higher when compared to PFPH.

The bioactive properties of proteins hydrolysates are tightly related to the nature of their amino acids (Maestri et al., 2018). Aromatic, hydrophobic, and positively charged amino acids were similar in both hydrolysates. The detailed amino acids composition of fenugreek protein hydrolysates is reported in Reference [16].

Potassium, sulphide, and phosphorus were the most abundant minerals in FPHs, while selenium and sodium concentrations represented the less abundant.

### 3.3. Antiproliferative Activity

We first examined if the exposition of Caco2 TC7 cells to 1 mg/mL of FPH inhibits their proliferation. The treatment of undifferentiated Caco2/TC7 cells with fenugreek proteins hydrolysates exhibited a decrease in their viability. The PFPH anti-proliferative property was time dependent and passed from 27% after 24 h to 55% after 72 h of the incubation period, compared to the control. With EFPH, there was also a cells proliferation inhibitory effect, which varied between 39% and 50%. Nevertheless, it was not significantly time dependent (Figure 2A). In order to demonstrate if the antiproliferative effect of FPH found on Caco2/TC7 cells was specific for the undifferentiated cells or was a cytotoxic mechanism, we tested this property on differentiated cells. There was no difference in differentiated cell growth between the control and the treated cells (Figure 2B).

### 3.4. Apoptosis Analysis

Two major mechanisms could lead to cell death: necrosis and apoptosis. Necrosis is characterized as passive, with uncontrolled release of inflammatory cellular contents. On the opposite side, apoptosis is considered to be a regulated and controlled process that avoids eliciting inflammation [24]. Thus, we examined which of the two mechanisms was triggered by FPH. After 24 h of incubation with PFPH and EFPH (final concentration 1 mg/mL) vs. untreated cells, undifferentiated Caco2 living cells decreased. Whereas, those with early apoptosis increased by 4.6-fold. There were no significant differences in cells with late apoptosis or necrosis before and after treatment (Figure 3).

### 3.5. Cell Cycle Analysis

We subsequently analyzed if the treatment with 1 mg/mL of FPH caused a cell-cycle arrest in Caco2 TC7. Cell-cycle analysis (Figure 4) showed that cells stopped in the G0-G1 phase were, respectively, 1.6-fold and 1.5-fold higher in PFPH and EFPH-treated cells compared to non-treated cells. In the S phase, the cells treated with EFPH, and not those treated with PFPH, decreased by 40% vs. the control cells. Even if there was a reduction in PFPH-treated cells blocked in the G2-M phase (−33%), this difference was not statistically significant.

### 3.6. Analysis of Mitochondrial Membrane Potential Change, Cytochrome C Release, and Caspase-3 Activation

Since FPH treatment (at 1 mg/mL) caused apoptosis in undifferentiated cells, we hypothesized that it could induce mitochondrial permeabilization and cytochrome C release in Caco-2/TC7 cells. Compared to non-treated cells, the number of cells exhibiting a changed in the mitochondrial membrane potential (ΔΨm) increased by 70% in PFPH-treated and EFPH-treated cells. The results also showed that, in treated cells, mitochondria cytochrome C contents decreased significantly compared to the untreated cells.

The release of cytochrome C can lead to the activation of caspase 3, which is an executor of the apoptosis pathway. The activated caspase-3 concentrations were significantly increased by 24-fold and 13-fold, respectively, in PFPH-treated and EFPH-treated cells when compared to the control (Table 2).

### 3.7. Antioxidant Activity of FPH in Caco2 Cells

Oxidative stress is a characteristic state of many cancers, and it is implicated in cancer development and progression. The intracellular ROS levels, in the presence of high concentration of H_2_O_2_ (20 mM), decreased by 35% in cells incubated with 1 mg/mL of EFPH when compared to the control. PFPH cells did not exhibit any modification. However, in the presence of low concentrations of H_2_O_2_ (0.5 mM), both treated cells exhibited better antioxidant activity vs. untreated cells. The inhibition reached 39% and 33%, respectively, in EFPH and PFPH treated cells (final concentration 1 mg/mL) (Figure 5).

### 3.8. Thioredoxin Reductase 1 Activity

Since FPH induced a decrease in intracellular ROS levels in Caco2 TC7, we proposed to study whether this decrease is caused by up-regulated enzyme activities or not. Hence, we proposed to measure the activity of one of the most important cellular antioxidant enzyme: thioredoxin reductase. TrxR1 activity was lower in PFPH (−41%) and EFPH (−12%) treated cells vs. control cells (Figure 6).

## 4. Discussion

The anticancer property of natural products became one of the most studied topics. In recent years, the studies on plant proteins and peptides have increased, which is motivated by their huge diversity, affordability, and lack of side effects. Legumes are the plant source for which most peptides with anticancer properties are reported [25]. Fenugreek is a legume-rich protein, which could be a potential source of biological active peptides.

The hydrolysis curve of fenugreek proteins was typical of many protein hydrolysates obtained by Sbroggio et al. [26] with okara hydrolysates. The differences in hydrolysis shape and DH values were probably due to the difference in enzyme specificity. On the other hand, the DH could inform the peptides’ mean size [17]. Hence, EFPH with DH = 19% could contain smaller peptides than PFPH.

Our results suggest that protein contents of the hydrolysates are important. These findings were in line with Pownall et al. [27] and Mundi and Aluko [28]. The high protein content could be a result of the solubilisation of peptides during hydrolysis. It is speculated that the hydrolysis, especially when alkaline enzymes are used, enhance the solubilisation of proteins and removes insoluble undigested non-protein substances [29].

Even if the amino acids profile showed that aromatic and hydrophobic amino acids did not differ between the hydrolysates, these values are higher than those found by other authors [27,28,30].

After analyzing the FPH composition, we tried to check if FPH possesses an anti-proliferative property in cells. The treatment of undifferentiated Caco2/TC7 cells with FPH exhibited a decrease in their viability, especially with PFPH that was correlated with incubation time. These results are in line with works reporting an anti-proliferative property of peptides and hydrolysates from soy [14], corn [12,31], chickpeas [13], and rice [11] on different cell models. Vglycin, a peptide isolated from soy, inhibited the proliferation of three types of colon cancer cells [14]. Ortiz-Martinez et al. [12] also found that corn peptide fractions decreased HepG2 cells growth by more than 50%. Li et al. [31] noticed that this antiproliferative property was time-dependent. In addition, a pentapeptide from rice brane showed 84% of viability inhibition on colon cancer cells [11].

Caco2/TC7 differentiated cell viability was not influenced by the FPH treatment. Same observation, in normal and cancer oral cells, was also reported by Kumar et al. [32] with a chickpea protein fraction. Ours findings could indicate a possible selective antiproliferative effect of PFPH and EFPH on cancer cells without affecting the normal cells.

One of the possible ways by which FPH inhibited the cancerous cells could be the same mechanism seen with antimicrobial peptides when they act as anticancer agents as well. It is believed that normal cells exhibit an asymmetric composition between the internal and the external layers of their membrane. In cancer cells, this asymmetry is affected principally by the externalization of phosphatidylserine (normally confined to the inner leaflet), and the external layer of cancer cell membranes that will carry a net negative charge. This permits an electrostatic interaction between cationic anticancer peptides and anionic cell membrane components [33].

It seemed interesting to investigate if FPH could induce the anti-proliferative effect by apoptosis. The study with EPFH and PEPH was carried out in undifferentiated cells by flow cytometry analysis after staining with annexin V/propidium iodide. Since cells in early apoptosis express phosphatidylserine in their outer side of the cytoplasmic membrane, they will be stained by Annexin-V labelled with FITC (early apoptosis). However, membranes of dead and damaged cells are permeable to propidium iodide (necrosis) and are also stained with annexin-V (late apoptosis) [20]. It seemed that PFPH and EFPH set off an early apoptosis mechanism, rather than necrosis, in undifferentiated cells. In this way, Ortiz-Martinez et al. [12] showed that HepG2 cells treated with corn peptide fractions have a four-fold increase of both early and late apoptotic events, compared to the untreated cells. Similarly, Li et al. [31] remarked that corn peptides generated apoptosis in 11% to 55% of HepG2 cells in a dose-dependent manner. Moreover, Vglycin treatment for 24 hours caused a significant increase of apoptosis in different colon cancer cells [14].

With apoptotic and no necrotic property, PFPH and EFPH seem to have a beneficial effect against cancer cells. Additionally, in this work, it was found that both FPH stimulated the early apoptosis, which is favored to the late one since it allows early recognition of dead cells [34].

By analyzing the effect of fenugreek proteins hydrolysates on cell cycles, it is suggested that both hydrolysates caused mainly a cell cycle arrest in the G1 phase, which has also been shown in other studies. In this way, Gao et al. [14] deduced that soy Vglycin induced a G1-phase arrest of colorectal cancer cells. Li et al. [31] indicated that corn peptides could induce HepG2 cell cycle arrest in the S phase. The hemagglutinin caused cell cycle arrest in the G2/M phase, as demonstrated by Lam and Ng [35].

Many studies have shown associations between some minerals and carcinogenesis. Mg^2+^ ions are enzyme cofactors involved in DNA repair mechanisms that maintain genomic stability and fidelity. Magnesium deficiency may also be associated with inflammation and increased levels of free radicals where both inflammatory mediators and free radicals arising could cause oxidative DNA damage and, therefore, tumor formation [36]. There is also evidence that dietary Ca^2+^ loading reduces colon cell proliferation and carcinogenesis [37]. The presence of these two elements in FPH could also be responsible for their anti-proliferative properties. According to Kasprzak [38], the molecular mechanisms involved in the effects of such minerals are likely to include binding at chromatin (e.g., DNA, histones, transcription factors, DNA repair enzymes) and other regulatory molecules in the target cells.

Apoptosis manifests in two major execution programs downstream of the death signal: the caspase pathway and organelle dysfunction of which mitochondrial dysfunction is best characterized [39]. To see if the apoptotic action of FPH was led by these mechanisms, we analyzed, by flow cytometry, the change in mitochondrial membrane potential, cytochrome C in the mitochondria, and the cytoplasmic level of the active form of caspase-3.

Mitochondria play a pivotal role in life and death of the cell since it produces the majority of energy required for survival and regulates the intrinsic pathway of apoptosis. The involvement of mitochondria in cell death is generally measured by following mitochondrial membrane depolarisation [21]. FPH-treated cells showed a higher change in mitochondrial membrane potential. Disruption of the mitochondrial outer membrane permeability leads to the release of proteins confined in the intermembrane space into the cytosol. These proteins include the apoptogenic factors, such as cytochrome C, which plays a crucial role in activating the mitochondrial-dependent death in the cytosol [32].

With the aim to discover whether PFPH and EFPH were able to induce mitochondrial permeabilization and cytochrome C release, we used flow cytometry to analyze the mitochondrial cytochrome C in treated and untreated cells. The results showed that, in treated cells, there was a greater cytochrome C release to cytoplasm than in the untreated cells. Once cytochrome C is released to the cytoplasm, it could activate different proteins of the intrinsic apoptosis pathway such as the effector caspase-3 [40]. Once caspase-3 is activated, it induces the proteolytic cleavage of a large number of essential proteins for apoptosis [41]. Moreover, caspase-3 is a prototypical executioner caspase that, upon activation by extrinsic and intrinsic pathways, cleaves a wide panel of several substrates that are vital for the cell, which precipitates regulated cell death. It is also responsible for modulating some enzyme activities like those required for the exposure of phosphatidylserine (PS) on the outer leaflet of dying cells [42].

Activated caspase-3 concentrations were increased in PFPH and EFPH treated cells. Gao et al. [14] also confirmed that Vglycin promoted caspase-3 activity in colon cancer cells. The same results were obtained by Li et al. [31], with corn peptides on HepG2 cells.

Higher levels of ROS are generated through the increased metabolic activity of cancer cells including enhanced signalling pathways or mitochondrial dysfunction [43]. The ROS levels in Caco2 cells were determined based on the reaction between ROS and DCFH-DA [13].

In our assays, Fenugreek protein hydrolysates showed antioxidant power. EFPH showed a better ROS inhibitory property even though PFPH was not effective with high levels of ROS. In this way, when HepG2 cells were incubated with 100 µM of peroxide, the corn peptides fraction could not decrease the peroxide-ROS generation [12]. In contrast, Xue et al. [13] showed that chickpea peptides decreased the ROS in MCF-7 and MDA-MB-231 cells. Torres-Fuentes et al. [44] and Zhang et al. [45] reported an antioxidant property of chickpea and soy proteins hydrolysates in Caco2 cells.

A study undertaken by Chi et al. [46] confirmed that peptides with a smaller molecular size, the presence of hydrophobic and aromatic amino acid residues, and the amino acid sequences were the key factors that determine the antioxidant activities of hydrolysates and peptides. Fenugreek protein hydrolysates are rich in hydrophobic and aromatic amino acids [16]. Moreover, as the cells incubated with protein hydrolysates were washed, some peptides may be lost since they are not able to cross the cell membrane due to their big size and polarity. However, small hydrophobic peptides are able to cross this membrane and stay in the cytoplasm, where they may exert their antioxidant property [44]. EFPH in which DH is higher than that of PFPH (19% vs. 9%, respectively) has higher ROS inhibition activity.

Because of the increase in ROS production in tumor cells, it is concerted that many antioxidants and redox control systems are up regulated. One of the most important cellular redox systems is the thioredoxin (Trx) system, comprised of Trx, TrxR1, and NADPH [43]. However, in our study, TrxR1 activity was found to be lower in both FPH-treated cells. Since the antioxidant enzymes activities are up-regulated following an increase of ROS production in cancer cells, we supposed that the decrease in TrxR1 could result from the low levels of ROS in treated cells (low stimuli of Trx and TrxR1 expression), and not a direct inhibition of the enzyme by FPH.

## 5. Conclusions

This data demonstrated that Purafect and Esperase fenugreek protein hydrolysates possess a selective antiproliferative property on colorectal cancer cells, by enhancing intrinsic apoptosis rather than necrosis on Caco2/TC7, and by blocking the cell cycle in the G1 phase. Both hydrolysates induced alteration in mitochondrial membrane permeability, induced cytochrome C release to the cytoplasm, and induced caspase-3 activation. Furthermore, these two hydrolysates exerted an antioxidant activity by inhibiting the reactive oxygen species. In light of these results, fenugreek proteins hydrolysates could represent a promising nutraceutical in the treatment and progression of colon cancer. Future studies will be interesting to perform in order to see if these fenugreek protein hydrolysates are also effective in other types of cancer cells and in vivo animal models.

## Figures and Tables

**Figure 1 nutrients-11-00724-f001:**
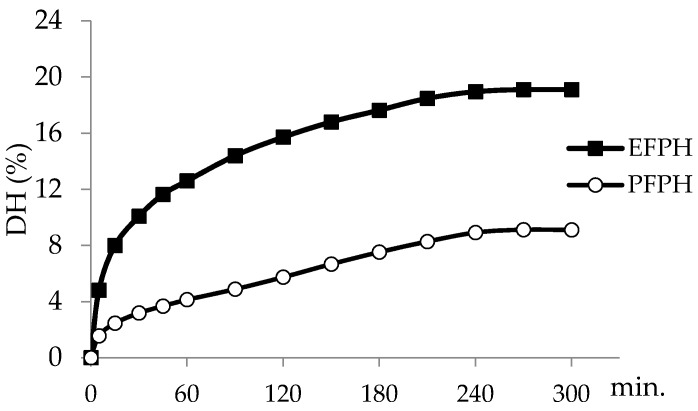
Kinetic of fenugreek proteins hydrolysis. E/S ratio= 5 U/mg proteins. EFPH: Esperase-fenugreek proteins hydrolysate. PFPH: Purafect-fenugreek proteins hydrolysate.

**Figure 2 nutrients-11-00724-f002:**
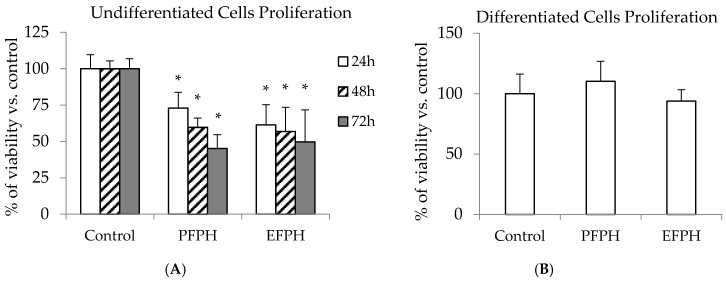
Relative viability of undifferentiated ((**A**): 24 h, 48 h, and 72 h) and differentiated Caco2/TC7 cells (**B**) treated (24 h) or not with fenugreek proteins hydrolysates. Data are presented as mean ± SD. The experiment was done in triplicate (each performed with six determinations). Superscripted (*) means are significantly different (*p* ≤ 0.05) compared to their respective control. Control: Untreated cells. PFPH: Purafect fenugreek proteins hydrolysate. EFPH: Esperase fenugreek proteins hydrolysate. The hydrolysates were used at a final concentration of 1 mg/mL.

**Figure 3 nutrients-11-00724-f003:**
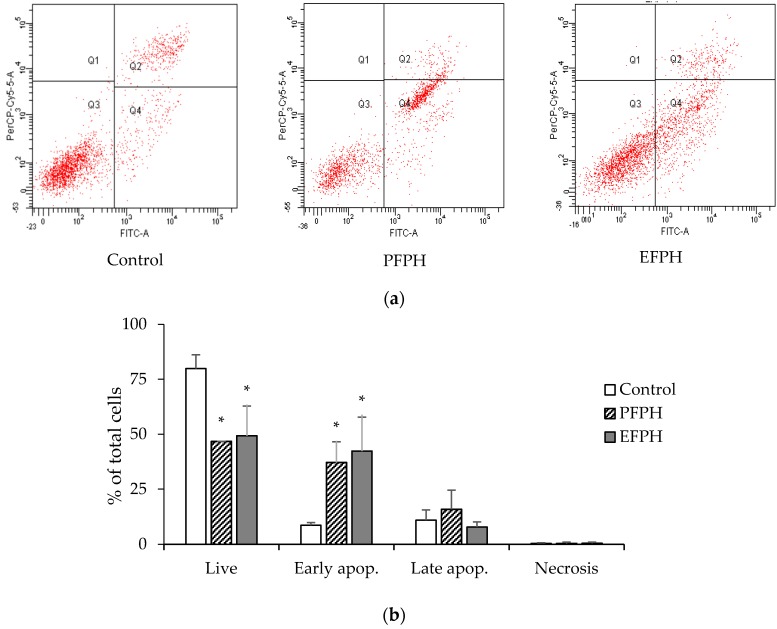
Effect of treating undifferentiated Caco2/TC7 cells with fenugreek proteins hydrolysates (24 h) on apoptosis. (**a**) Representative histogram of cytometry analysis. (**b**) Cell death process repartition. Data are presented as mean ± SD. The experiment was done in duplicate. Superscripted (*) means are significantly different (*p* ≤ 0.05) when compared to their respective control. Control: Untreated cells. PFPH: Purafect fenugreek proteins hydrolysate. EFPH: Esperase fenugreek proteins hydrolysate. The hydrolysates were used at a final concentration of 1 mg/mL.

**Figure 4 nutrients-11-00724-f004:**
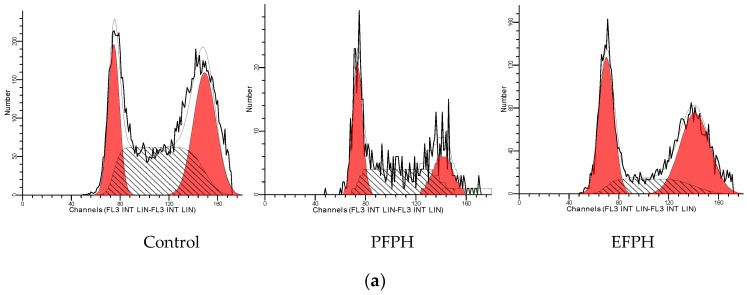
Cell cycle repartition of undifferentiated Caco2/TC7 treated (24 h) or not with fenugreek proteins hydrolysates. (**a**): a representative cells cycle histogram. (**b**): G1, S, and G2 phases percentage distribution. Data are presented as mean ± SD. The experiment was done in duplicate. Superscripted (*) means are significantly different (*p* ≤ 0.05) when compared to their respective control. Control: Untreated cells. PFPH: Purafect fenugreek proteins hydrolysate. EFPH: Esperase fenugreek proteins hydrolysate. The hydrolysates were used at a final concentration of 1 mg/mL.

**Figure 5 nutrients-11-00724-f005:**
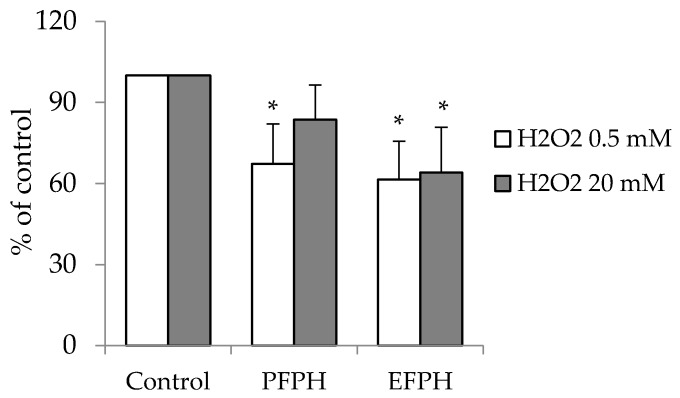
Relative reactive oxygen species levels in undifferentiated Caco2/TC7 cells treated (24 h) or not with fenugreek proteins hydrolysates. Data are presented as mean ± SD. The experiment was done in triplicate (each performed with six determinations). Superscripted (*) means are significantly different (*p* ≤ 0.05) compared to their respective control. Control: Untreated cells. PFPH: Purafect fenugreek proteins hydrolysate. EFPH: Esperase fenugreek proteins hydrolysate. The hydrolysates were used at a final concentration of 1 mg/mL.

**Figure 6 nutrients-11-00724-f006:**
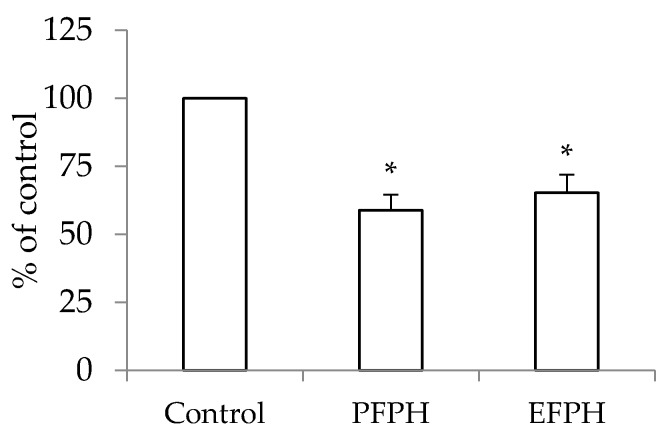
Relative thioredoxin reductase activity in undifferentiated Caco2/TC7 cells treated (24 h) or not with fenugreek proteins hydrolysates. Data are presented as mean ± SD. The experiment was done in duplicate (each performed with six determinations). Superscripted (*) means are significantly different (*p* ≤ 0.05) compared to their respective control. Control: Untreated cells. PFPH: Purafect fenugreek proteins hydrolysate. EFPH: Esperase fenugreek proteins hydrolysate. The hydrolysates were used at a final concentration of 1 mg/mL.

**Table 1 nutrients-11-00724-t001:** Chemical composition of fenugreek protein hydrolysates.

	PFPH	EFPH
	(%)	(%)
Proteins	89.9 ± 0.2	92.3 ± 0.5 *
HAA	49.4	49.0
AAA	16.8	17.2
PCAA	12.2	12.5
Lipids	2.8 ± 0.2	3.3 ± 0.4 *
Total fiber	3.0 ± 0.1	2.0 ± 0.1 *
Carbohydrates ^#^	1.5	˂1
Moisture	1.0 ± 0.1	1.1 ± 0.1
Ash	1.8 ± 0.2	1.3 ± 0.3
Mineral composition		
Potassium (mg/100 g)	986 ± 5	1001 ± 1 *
Phosphorus (µg/g)	1933 ± 7	1927 ± 11
Sulphide (µg/g)	1128 ± 5	1268 ± 48 *
Magnesium (µg/g)	1013 ± 4	985 ± 9 *
Calcium (µg/g)	636 ± 2	502 ± 14 *
Sodium (µg/g)	122 ± 4	110 ± 3 *
Selenium (µg/kg)	53 ± 1	52 ± 1

PFPH: Purafect fenugreek proteins hydrolysate. EFPH: Esperase fenugreek proteins hydrolysate. HAA: hydrophobic amino acids (Ala, Val, Ile, Leu, Tyr, Phe, Trp, Pro, Met, and Cys). AAA: aromatic amino acids (Phe, Tyr, Trp). PCAA: positively charged amino acids (Arg, His, Lys). Results are presented as mean ± SD (*n* = 3). Superscripted (*) means within a row are significantly different (*p* ≤ 0.05). #: Calculated by difference.

**Table 2 nutrients-11-00724-t002:** Percentage of Caco2/CT7 cells with a positive mitochondrial membrane potential. Cells with mitochondrial cytochrome C and active caspase-3, quantified by flow cytometry in response to fenugreek proteins hydrolysates treatment (24 h).

Number of Cells (/100 Cells)	Control	PFPH	EFPH
Cells with positive MMP	28.7 ± 10.9	69.2 ± 11.4 *	70.4 ± 4.1 *
Cells with mitochondrial cytochrome C	90.7 ± 1.6	71.3 ± 1.2 *	61.0 ± 9.8 *
Cells with active caspase-3	1.4 ± 0.0	33.3 ± 4.3 *	18.5 ± 3.8 *

Data are presented as mean ± SD. The experiment was done in duplicate. Superscripted (*) means within a row are significantly different (*p* ≤ 0.05) when compared to their respective control. Control: untreated cells. PFPH: Purafect fenugreek proteins hydrolysate. EFPH: Esperase fenugreek proteins hydrolysate. The hydrolysates were used at a final concentration of 1 mg/mL. MMP: mitochondrial membrane potential.

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
