# Peer review of "Protein Hydrolysates from Fenugreek (Trigonella foenum graecum) as Nutraceutical Molecules in Colon Cancer Treatment"

_nutrients, 2019, doi:10.3390/nu11040724_

Round 1
Reviewer 1 Report
Allaoui et al., investigate if protein hydrolysates from fenugreek can be use as supplement in the treatment of colon cancer. The authors evaluated the effect of hydrolysates on cell proliferation, apoptosis, cell cycle and antioxidant activity in a colon cancer cell line. The results show a clear effect on cell proliferation in particular with one of the hydrolysates, this decrease in proliferation could be a consequence of a combination of apoptosis and cell cycle arrest. The authors conclude that the hydrolysates show promising nutraceutical in the treatment of colon cancer.
The data present in this paper is interesting and another evidence that natural plants can be used as supplement for several diseases. The methodology used supports the conclusion that these hydrolysates have an effect on the proliferation colon cancer cell line. The authors propose further studies by evaluating other types of cancer and in vivo experiments.
Mayor revisions
The paper will benefit with a change in the title, the aim of the paper is to demonstrate the hydrolysates can help in the treatment of colon cancer; my suggestion will be to change to "Protein hydrolysates from fenugreek (Trigonella frenum graecum) as nutraceutical molecules in colon cancer treatment".
The paper will be better presented if in each of the results chapters, the authors start explaining the aim of the experiment and finishing with a general conclusion. As currently presented, the authors just describe the data without the reader understanding why that experiment was performed.
Those this effects are exclusive of this colon cancer cell line or other colon cancer cell lines behave in similar ways.
The discussion includes paragraphs that cover what I refer in 2. I suggest the re-arrangement of these paragraphs to the results and a more concise discussion.
Minor revisions.
Line 38 in the introduction, the authors are enlisting the diseases were plants have been proven to be beneficial, at the end of the phrase the authors added dots and etcetera. the phrase should be ended with digestive disorders and period. After that references about all of this studies should be listed.
Some of the materials and methods will improve with a small in brief explanation than the just references.
What is the difference between undifferentiated and differentiated cells? What are the molecular differences and their consequences in cancer progression?
Line 118 and 127 the correct terminology is fetal bovine serum
Line 168, the concentration of 500 M H2O2 is missing.
The reader will benefit if the concentration of the hydrolysates was mentioned in the results text and in the figures legends. Although is describe in material and methods it will help that the reader doesn't have to look back for the information.
Figures and text will benefit if the authors spell control or vehicle in their graphs and in the results text.
Line 219, legend in Table 1. The authors mentioned that the detailed amino acids composition of.... is reported in 16; that phrase should be in the result text no in the table.
Figure 3, vertical axis of the graph is labeled as %. Authors need to clarify it its percent of live or dead.
Figure 4, again graph is labeled as %. Clarify if its % of total cells, and how many events were considered for this quantification.
The authors correctly present references as: first author, et al.; except in two references in the discussion were they named all the authors of the paper: Ortiz-Martinez, de Mejia, Garcia-Lara...... and Li, Zhang, He, Ma...; they should keep the format as first author, et al.
Author Response
Answers to referee 1
Allaoui et al., investigate if protein hydrolysates from fenugreek can be use as supplement in the treatment of colon cancer. The authors evaluated the effect of hydrolysates on cell proliferation, apoptosis, cell cycle and antioxidant activity in a colon cancer cell line. The results show a clear effect on cell proliferation in particular with one of the hydrolysates, this decrease in proliferation could be a consequence of a combination of apoptosis and cell cycle arrest. The authors conclude that the hydrolysates show promising nutraceutical in the treatment of colon cancer.
The data present in this paper is interesting and another evidence that natural plants can be used as supplement for several diseases. The methodology used supports the conclusion that these hydrolysates have an effect on the proliferation colon cancer cell line. The authors propose further studies by evaluating other types of cancer and in vivo experiments.
Mayor revisions
The paper will benefit with a change in the title, the aim of the paper is to demonstrate the hydrolysates can help in the treatment of colon cancer; my suggestion will be to change to "Protein hydrolysates from fenugreek (Trigonella frenum graecum) as nutraceutical molecules in colon cancer treatment".
The title has been changed as suggested by the reviewer.
The paper will be better presented if in each of the results chapters, the authors start explaining the aim of the experiment and finishing with a general conclusion. As currently presented, the authors just describe the data without the reader understanding why that experiment was performed.
This suggestion has been taken into account and the changes have been made.
Those this effects are exclusive of this colon cancer cell line or other colon cancer cell lines behave in similar ways.
In the present work, the anticancer properties of fenugreek hydrolysates were evaluated on Caco2 cells TC7 line, so the conclusions were made on this type of cells. These hydrolysates could have the same effect on other lines; however, this should be confirmed by other studies. For this purpose, we suggested, in the end of conclusion, to undertake other studies to examine if these hydrolysates are also effective in other types of cancer cells, including other lines of colon cancer.
The discussion includes paragraphs that cover what I refer in 2. I suggest the re-arrangement of these paragraphs to the results and a more concise discussion.
The authors have transferred some of these paragraphs from the discussion to the results.
Minor revisions.
Line 38 in the introduction, the authors are enlisting the diseases were plants have been proven to be beneficial, at the end of the phrase the authors added dots and etcetera. the phrase should be ended with digestive disorders and period. After that references about all of this studies should be listed.
The indicated paragraph was synthetized from a review which listed a table with medicinal uses of fenugreek in some diseases and health problems. The reference of this review has been added to the paragraph.
Some of the materials and methods will improve with a small in brief explanation than the just references.
Methods in sections from 2.6 to 2.10 have been detailed as proposed by the reviewer.
What is the difference between undifferentiated and differentiated cells? What are the molecular differences and their consequences in cancer progression?
Undifferentiated cells are characterized by a rapid and uncontrolled division, which is identical to the cancerous cells. These cells exhibit a high expression and activity of anti-apoptotic molecules, and low activity of pro-apoptotic molecules. Whereas, differentiated cells represent normal cells, with controlled division.
Caco-2 cell line, derived from human colon adenocarcinoma, is one of the most regulatory used for intestinal drug studies. It undergoes spontaneous enterocytes differentiation when cultured over confluence for 21 days to become polarized cells expressing apical and basolateral surfaces with well-established tight junctions.
Chantret et al. J. Cell Sci. 1994, 107: 213-25.
Cell confluence (80%) is confirmed by microscopic observance. The experiments were carried out in cancerous or undifferentiated cells (24 hours post-seeding to prevent cell differentiation).
Line 118 and 127 the correct terminology is fetal bovine serum
The word foetal in “foetal bovine serum” has been corrected by fetal.
Line 168, the concentration of 500 M H2O2 is missing.
The unit of the corresponding concentration has been corrected and is 500 µM.
The reader will benefit if the concentration of the hydrolysates was mentioned in the results text and in the figures legends. Although is describe in material and methods it will help that the reader doesn't have to look back for the information.
As recommended by the reviewer, the concentration of the hydrolysates has been added in figures and table 3 footnotes, and also in the results section.
Figures and text will benefit if the authors spell control or vehicle in their graphs and in the results text.
The abbreviation C has been spelled as “control” in the main text, figures and table 3.
Line 219, legend in Table 1. The authors mentioned that the detailed amino acids composition of.... is reported in 16; that phrase should be in the result text no in the table.
The Phrase “The detailed amino acids composition of fenugreek proteins hydrolysates is reported in [16]” has been removed from footnote in table 1, and has been replaced in paragraph 3.2. after amino acids description.
Figure 3, vertical axis of the graph is labeled as %. Authors need to clarify it its percent of live or dead.
The percentage in the y-coordinate represents the percent of total cells. This information has been added to the graph.
Figure 4, again graph is labeled as %. Clarify if its % of total cells, and how many events were considered for this quantification.
Here again, the percentage in the y-coordinate represents the percent of total cells. This information has been added to the graph.
Cycle events per channel was 179, and All cycle events was 12307.
The authors correctly present references as: first author, et al.; except in two references in the discussion were they named all the authors of the paper: Ortiz-Martinez, de Mejia, Garcia-Lara...... and Li, Zhang, He, Ma...; they should keep the format as first author, et al.
The references have been checked and all those badly done have been corrected. Those references have been highlighted in yellow in the text.
Reviewer 2 Report
Suggestion 1: Line 155-160 can be replaced by brief description of caspase 3 and cytochrome C detection methods.
Suggestion 2: Table 1 revealed the mineral composition differences of PFPH and EFPH. Besides, Figure 4 demonstrated that EFPH is a effective DNA synthesis suppressor than PFPH. Therefore, to discuss the role of minerals in regulation of colon cancer cell cycle will be an interesting point of view.
Author Response
Answers to referee 2
Comments and Suggestions for Authors
Suggestion 1: Line 155-160 can be replaced by brief description of caspase 3 and cytochrome C detection methods.
The detailed methods for caspase 3 and cytochrome C analysis have been added as suggested by the reviewer.
Suggestion 2: Table 1 revealed the mineral composition differences of PFPH and EFPH. Besides, Figure 4 demonstrated that EFPH is a effective DNA synthesis suppressor than PFPH. Therefore, to discuss the role of minerals in regulation of colon cancer cell cycle will be an interesting point of view.
As proposed by the reviewer, a new paragraph has been added in line 428, and the new references have been added in the list of references (highlighted in yellow):
“Many studies have evidenced associations between some minerals and carcinogenesis. Indeed, Mg2+ ions are enzyme cofactors involved in DNA repair mechanisms that maintain genomic stability and fidelity. Magnesium deficiency may also be associated with inflammation and increased levels of free radicals where both inflammatory mediators and free radicals so arising could cause oxidative DNA damage and therefore tumour formation [37]. There is also much evidence that dietary Ca2+ loading reduces colon cell proliferation and carcinogenesis [38]. The presence of these two elements in FPH could also be responsible of their antiproliferative properties. According to Kasprzak [39], the molecular mechanisms involved in the effects of such minerals are likely to include binding at chromatin (e.g., DNA, histones, transcription factors, DNA repair enzymes) and other regulatory molecules in the target cells.”
However, in this text only Mg+2 and Ca+2 have been discussed, as the significant relationship of sodium to gastric carcinogenesis is not proved, and sulphides effect on carcinogenesis is controversial. Selenium and phosphorus were not significantly different between FPH.